# Ferroptosis Inducers Erastin and RSL3 Enhance Adriamycin and Topotecan Sensitivity in ABCB1/ABCG2-Expressing Tumor Cells

**DOI:** 10.3390/ijms26020635

**Published:** 2025-01-14

**Authors:** Lalith Perera, Shalyn M. Brown, Brian B. Silver, Erik J. Tokar, Birandra K. Sinha

**Affiliations:** 1Laboratory of Genome Integrity and Structural Biology, National Institutes of Environmental Health Sciences (NIH), Research Triangle Park, North Carolina, NC 27709, USA; lalith.perera@nih.gov; 2Mechanistic Toxicology Branch, Division of Translational Toxicology, National Institutes of Environmental Health Sciences (NIH), Research Triangle Park, Durham, NC 27709, USA; shalyn.brown@nih.gov (S.M.B.); brian.silver@nih.gov (B.B.S.); erik.tokar@nih.gov (E.J.T.); 3Boonshoft School of Medicine, Wright State University, Dayton, OH 45435, USA

**Keywords:** P-gp protein, breast cancer resistance protein, Adriamycin, Topotecan, ferroptosis, Erastin, RSL3

## Abstract

Acquired resistance to chemotherapeutic drugs is the primary cause of treatment failure in the clinic. While multiple factors contribute to this resistance, increased expression of ABC transporters—such as P-glycoprotein (P-gp), breast cancer resistance protein (BCRP), and multidrug resistance proteins—play significant roles in the development of resistance to various chemotherapeutics. We found that Erastin, a ferroptosis inducer, was significantly cytotoxic to NCI/ADR-RES, a P-gp-expressing human ovarian cancer cell line. Here, we examined the effects of both Erastin and RSL3 (Ras-Selected Ligand 3) on reversing Adriamycin resistance in these cell lines. Our results show that Erastin significantly enhanced Adriamycin uptake in NCI/ADR-RES cells without affecting sensitive cells. Furthermore, we observed that Erastin enhanced Adriamycin cytotoxicity in a time-dependent manner. The selective iNOS inhibitor, 1400W, reduced both uptake and cytotoxicity of Adriamycin in P-gp-expressing NCI/ADR-RES cells only. These findings were also confirmed in a BCRP-expressing human breast cancer cell line (MCF-7/MXR), which was selected for resistance to Mitoxantrone. Both Erastin and RSL3 were found to be cytotoxic to MCF-7/MXR cells. Erastin significantly enhanced the uptake of Hoechst dye, a well-characterized BCRP substrate, sensitizing MCF-7/MXR cells to Topotecan. The effect of Erastin was inhibited by 1400W, indicating that iNOS is involved in Erastin-mediated enhancement of Topotecan cytotoxicity. RSL3 also significantly increased Topotecan cytotoxicity. Our findings—demonstrating increased cytotoxicity of Adriamycin and Topotecan in P-gp- and BCRP-expressing cells—suggest that ferroptosis inducers may be highly valuable in combination with other chemotherapeutics to manage patients’ cancer burden in the clinical setting.

## 1. Introduction

Chemotherapy drug resistance, especially multidrug resistance (MDR), poses significant challenges in cancer treatment. Key problems associated with MDR include genetic mutation [1], tumor heterogeneity [2], enhanced repair of DNA damage [3,4], and increased efflux of drugs via ABC transporters expressed in tumor cells [5]. Overexpression of ATP-binding cassette (ABC) transporters, such as P-glycoprotein (P-gp, ABCB1), BCRP (Breast Cancer Resistant Protein, ABCG2), and MRP (Multidrug Resistant Protein, ABCC1), actively transport a wide range of drugs out of cells, leading to reduced drug accumulation and diminishing therapeutic efficacy [5,6,7]. Because many of the clinically active anticancer drugs currently used in the clinic are substrates of ABC transporters, treatment failure often results in poor prognosis and ultimately increased mortality. MDR not only limits effectiveness to one class of drugs but to a broad range of anticancer drugs complicating treatment by causing resistance to unrelated drugs. Consequently, overcoming resistance requires higher doses of drugs, leading to increased toxicity and adverse effects for patients. 

While MDR is a leading cause of therapeutic failure and relapse in cancer patients, recent research suggest that specific inhibitors of ABC transporters can be an effective approach to overcome resistance [8,9]. It is expected that combining inhibitors with chemotherapeutic agents will result in enhanced drug accumulation within tumor cells with improved treatment efficacy. It is also possible to use various single agents and combination therapies to target multiple pathways simultaneously, to increase efficacy with fewer side effects.

Recently, ferroptosis inducers are being recognized as a significant step forward for the treatment of many cancers and to overcome drug resistance [10,11,12]. Ferroptosis is a form of oxidative cell death which is morphologically distinct from apoptosis-associated characteristics such as cellular shrinkage and plasma membrane blistering [13,14]. Ferroptosis is an iron-dependent form of non-apoptotic cell death induced by Erastin, a ferroptosis inducer, triggered by excess reactive oxygen species (ROS) generation and lipid peroxidation [15,16]. Mechanistically, ferroptosis is associated with inhibition of glutathione peroxidase 4 (GPX4), an enzyme involved in the reduction and formation of lipid peroxides. While RSL3 is a direct inhibitor of GPX4 inducing ferroptosis [17], ER induces ferroptosis by the inhibition of the amino acid antiporter system Xc^−^ by blocking the influx of extracellular cystine [11]. The depletion of intracellular cysteine leads to a reduced GSH biosynthesis, resulting in the depletion of GSH, a reducing co-substrate for many cellular proteins, including GPX4. The inhibition of GPX4 leads to the accumulation of LOS, lipid peroxidation and ultimately ferroptotic cell death.

Ferroptosis inducers, like ER and RSL3, have shown promise in sensitizing resistant tumor cells to various chemotherapeutic agents [18,19] and have emerged as promising candidates for addressing MDR. Cancer cells are more vulnerable to iron toxicity due to high dependency on iron metabolism than noncancerous cells, making cancer cells more vulnerable to ferroptosis. Combining ferroptosis inducers with chemotherapeutics offers a novel strategy to enhance treatment efficacy and overcome resistance.

In this study, we have evaluated the role of ER and RSL3 in sensitizing both ABCB1- and ABCG2-expressing human tumor cells to Adriamycin and Topotecan. Our finding shows that ER and RSL3 play a critical role in overcoming drug resistance in human ovarian and breast tumors cells through their ability to generate nitric oxide (^●^NO) and inhibit GPX4, respectively. These events (^●^NO formation and inhibition of GPX4) eventually lead to cellular accumulation of ^●^NO, ROS, and lipid-ROS, ultimately causing ferroptotic cell death.

## 2. Results

### 2.1. Cytotoxicity of Erastin and RSL3 in Ovarian and Breast Tumor Cells

As previously shown, Pg-p-expressing cells (NCI/ADR-RES) were extremely resistant to Adriamycin (ADR) compared to OVCAR-8 cells [20]. Similarly the BCRP-expressing cells, MCF-7/MXR, were highly resistant to Topotecan (TPT) compared to MCF-7 cells [21]. While ER has been shown to be a substrate of P-gp in certain P-gp-expressing tumor cells [22], in this study, NCI/ADR-RES cells showed no resistance to ER or RSL3 (Figure 1A,C). While ER may also be a substrate for BCRP [23], our studies indicated that BCRP-expressing MCF-7/MXR cells were more sensitive to both ER and RSL3 compared to the parent MCF-7 cells (Figure 1B,D), suggesting that ER was not a substrate for BCRP in MCF-7/MXR cells.

### 2.2. Sensitization of Adriamycin by Erastin and RSL3 in Ovarian Cancer Cells

To examine the ability of ER and RSL3 to overcome ADR resistance in NCI/ADR-RES cells, we treated these cells with ER at different times. Data presented in Figure 2A,B shows that these inducers are indeed able to sensitize NCI/ADR-RES cells to ADR, albeit only slightly by ER without significantly affecting the parent OVCAR-8 cells. Furthermore, we found that longer incubations with ER at lower concentrations resulted in significantly higher sensitization to ADR, aligning with our previous findings [24]. 1400W, a selective inhibitor of iNOS [25], significantly affected cytotoxicity of ADR in NCI/ADR/RES cells, suggesting that ^●^NO or similar species are involved. In contrast to ER, RSL3 was more effective in sensitizing NCI/ADR-RES cells to ADR (Figure 2C). In addition, RSL3 was also effective in sensitizing WT cells to ADR and TPT (Figure 2D,E).

### 2.3. Sensitization of Topotecan by Erastin and RSL3 in Breast Cancer Cells

ER enhanced TPT cytotoxicity in MCF-7/MXR cells (Figure 3A,B) without significantly affecting the parent MCF-7 cells (Figure 3D). Intriguingly, the iNOS inhibitor, 1400W, significantly decreased ER-induced cytotoxicity of TPT in MXR cells (Figure 3), suggesting that ^●^NO or similar species are involved in ER-mediated sensitization of TPT in MXR cells. RSL3 was effective in both MCF-7 and MCF-7/MXR cells (Figure 3C,E).

### 2.4. Erastin Enhances Uptake of Adriamycin in Ovarian NCI/ADR-RES Cells

To better understand mechanisms of ER-enhanced cytotoxicity of ADR in NCI-ADR-RES cells, the effects of ER on uptake/retention of ADR were examined in this cell line. Our results (Figure 4 and Appendix A) showed that the pretreatment of cells with ER results in increased accumulation of ADR in the resistant cells. In contrast, ER did not significantly increase ADR uptake in the sensitive OV-WT cells (Figure 4A). The effects of 1400W were also investigated as 1400W inhibited ER-induced uptake and increased ADR cytotoxicity into these cells. 1400W was effective in inhibiting the effects of ER (Figure 4B).

### 2.5. Erastin Enhances Uptake of Hoechst Dye in MCF-7/MXR Breast Cells

We investigated whether ER also increased Topotecan uptake/retention in MCF-7 and MCF-7/MXR cells for enhanced cytotoxicity. We utilized Hoechst dye, a well-known and well-studied substrate for BCRP [26,27]. Our results depicted in Figure 5 (and Appendix A) show that ER significantly enhanced the uptake of Hoechst dye in MCF-7/MXR cells without significantly modulating the uptake of the dye in MCF-7 cells (Figure 5A). 1400W failed to inhibit the ER-mediated uptake of the dye in these resistant cells; in fact, 1400W enhanced the dye uptake. The reason for this observation is unclear, but it may suggest that 1400W requires a much longer incubation to be effective as seen in cytotoxicity studies.

Figure 6A,B (and Appendix A) show that ER did not affect the expression of either P-gp or BCRP proteins, neither over time nor in concentrations. This indicates that ER does not inhibit translation or change degradation (steady state/half-life) of the ABC-transporter proteins in these cell lines.

### 2.6. Molecular Modeling

ER has been shown to bind to P-gp and has been shown to act as a weak inhibitor/substrate for the protein. Docking analysis was conducted to investigate the binding interactions of Erastin (ER) with P-glycoprotein (P-gp, ABCB1) and Breast Cancer Resistance Protein (BCRP, ABCG2). We focused our docking studies on the Taxol-binding sites of P-gp and the camptothecin-binding sites of BCRP, as these are critical for their respective transport functions. Furthermore, we carried out extensive energy calculations of ADR and TPT to these binding sites and compared them with ER binding (Table 1 and Table 2). Our results clearly showed that ER binds very efficiently to both P-gp-170 and BCRP with high binding affinity (Table 1 and Table 2). Furthermore, ER also binds to similar binding sites as Adriamycin in P-gp (Figure 7) and TPT in BCRP (Figure 8). Our studies also showed that ER formed significant hydrogen bonds with residues Y310 and Q725 in P-gp while also engaging in hydrophobic interactions with F978 and I980. These interactions suggest a stable binding mode that may influence ATPase activity, consistent with the observed increase in Adriamycin uptake. Erastin exhibited hydrogen bonding with S596 and Q141 and hydrophobic interactions with L539 and V534 in BCRP. These binding interactions align with the increased Hoechst dye uptake observed in BCRP-expressing cells treated with Erastin.

## 3. Discussion

Acquired resistance to chemotherapy is the leading cause of treatment failure in the clinic today [28,29,30]. Among the mechanisms involved for the emergence of drug resistance, overexpression of ABC transporters, e.g., P-170 glycoprotein (P-gp, MDR1, ABCB1), breast cancer resistant protein (BCRP, ABCG2) and multi-resistance proteins (MRP’s) play a significant role in mediating resistance by actively removing various anticancer drugs from cancer cells. Although considerable success has been achieved in inhibiting the activity of ABC transporters in vitro, currently there are no compounds that have successfully inhibited/blocked ABC transporter-mediated resistance in the clinic.

Ferroptosis inducers especially ER and RSL3 have been reported to sensitize certain tumor cells to chemotherapeutics [18,19,31]. Zhou et al. [22] have shown that ER enhances cytotoxic effects of Docetaxel against ABCB1-expressing ovarian tumor cells by inhibiting efflux of the drug. Interestingly, in this study ER was found to be a substrate of ABCB1 in the Taxol-selected resistant ovarian tumor cells. In contrast, in our model cell line, ABCB1-expressing ovarian NCI/ADR-RES cells, ER was similarly cytotoxic to both sensitive OVCAR-8 and ABCB1-expressing resistant variant, suggesting that ER was not a substrate of P-170 glycoprotein. Additionally, we found that ER was not a substrate of BCRP as ER was more active against the BCRP-overexpressing MCF-7/MXR cells than the WT-MCF-7 cells. 

Our studies show that the ferroptosis inducers, ER and RSL3, enhanced ADR and TPT cytotoxicity in both P-gp-expressing NCI/ADR-RES and BCRP-expressing MCF-7/MXR cells, respectively. We found that ER increased cellular uptake of both ADR and Hoechst dye in these resistant cell lines without significantly modulating the uptake of ADR or Hoechst dye in sensitive OV-WT or MCF-7 cells. Our studies also show that 1400W, a selective inhibitor of iNOS, modulated both the uptake and cytotoxicity of ADR and TPT in these cell lines. Again, 1400W had no significant effects on the uptake or the cytotoxicity of ADR or TPT in sensitive WT cells. Our observations suggest that ^●^NO or related species may facilitate ER cytotoxicity in the resistant cells. This aligns with our earlier studies showing that ^●^NO inhibits ATPase activity [32], including the ABC transporters [21,33], and potentially increasing the uptake and retention of drugs for enhanced cytotoxicity. Several studies have reported the formation of ^●^NO in various cancer cell lines following treatment with ER, resulting in ferroptotic cell death [34,35]. Hou et al. [35] have reported that this ER-induced activation of PDI results in dimerization of nNOS and accumulation of cellular ^●^NO, ROS and lipid ROS, resulting in ferroptotic cell death in immortalized HT22 mouse hippocampal neuronal cells. NCX4040, a ^●^NO-generating drug, induces ferroptosis in colon cancer cells which was further enhanced by ER [36].

In a separate study, we found that ER elicits not only ferroptosis but also other forms of cell death which contribute to growth inhibition of MCF-7 cells (manuscript in preparation). In addition to exhibiting high antitumor activity as a single agent at reasonably low concentrations, ER enhances the sensitivity of both NCI/ADR-RES and MCF-7/MXR cells to Adriamycin and Topotecan, respectively. Furthermore, we show that RSL3 was notably more effective as a single agent against both sensitive and resistant cells, albeit resistant cells were significantly more sensitive to RSL3, indicating that RSL3 exerts its effects independently of ABC transporters, most likely by inhibiting GPX4.

These findings, therefore, provide a promising avenue for combination therapies in cancer treatment, especially for patients harboring ABC transporter-expressing tumors. Ferroptosis inducer-dependent oxidative stress and lipid peroxidation may bypass protective effects of ABC transporters and lead to sensitization of MDR tumors to chemotherapeutic agents. Molecular modeling indicated strong binding affinities of ER to both ABCB1 and ABCG2 proteins, supporting the hypothesis that Erastin interacts directly with ABC transporters, modulating their activity and thereby enhancing chemotherapeutic drug retention in resistant tumor cells. While the docking results provide valuable insights into the potential binding modes and affinities of Erastin with P-gp and BCRP, there are some limitations to this as docking studies consider proteins as rigid or semi-flexible structures, which do not account for their dynamic conformational flexibility, potentially overlooking alternative binding orientations. Molecular dynamic (MD) simulations may offer a more comprehensive understanding of time-dependent changes in ligand–protein complexes. MD simulations could also provide insights into the stability of Erastin’s binding under physiological conditions, revealing conformational changes in P-gp and BCRP upon ligand binding.

Additionally, our studies show that ^●^NO is generated in resistant cells, resulting in inhibition of ATPase functions, reducing drug efflux and enhancing cytotoxicity of chemotherapeutics. While these are promising findings, additional research is needed to fully understand the relationship between ER-induced ^●^NO formation, iNOS, and ABC transporters in vivo and other tumor models. Additional research is needed to address potential limitations, including off-target effects and toxicity profiles of ferroptosis inducers. This would further clarify the clinical potentials for combining ferroptosis inducers with chemotherapeutic agents as novel combination therapies to overcome drug resistance in patients.

## 4. Materials and Methods

Adriamycin (ADR, >98%) was the gift of the Drug Synthesis and Chemistry Branch, Developmental Therapeutic Program of NCI, NIH. ADR was dissolved in double-distilled water and stored at −80 °C. Topotecan HCl (TPT, >98%), Erastin (>98%), and RSL3 (>98%) were purchased from Cayman Chemicals (Ann Arbor, MI, USA). Stock solutions of TPT, ER, and RSL3 were prepared in in DMSO and stored at −80 °C.

### 4.1. Cell Culture

Human ovarian OVCAR-8 and NCI/ADR-RES cell lines were obtained from the NCI-Frederick Cancer Center (Frederick, MD, USA), and human MCF-7 breast tumor cells were obtained from ATCC (Manassas, VA, USA). MCF-7/MX breast tumor cells (MXR), selected for resistance by exposure to Mitoxantrone as described before [37], was a gift of Doctor Erasmus Schneider (NCI/NIH). Cells were cultured in Phenol Red-free RPMI media, supplemented with 10% fetal bovine serum and antibiotics. Cells were routinely used for 20–25 passages, after which the cells were discarded, and a new cell culture was started from fresh, frozen stock. 

### 4.2. Cytotoxicity Studies

Cytotoxicity was assessed by both CellTiter-Glo (Promega, Madison, WI, USA) and Trypan Exclusion methods. For the CellTiterGlO assay, 2500–3000 cells/well were seeded in opaque white, 96-well plates and allowed to attach overnight. Cells were then treated with various concentrations of drugs and incubated for 72 h, and cytotoxicity was determined according to the manufacturer’s instructions. For the trypan blue assay, about 25,000–50,000 cells/wells were seeded onto a 12-well plate (in duplicates), allowed to attach for 18 h and treated with various drugs for 24 h or 72 h. Following trypsinization, surviving cells were collected, and 15 µL of cell mixtures were combined with 15 µL of trypan blue and counted in a T20 automatic cell counter (Bio-Rad, Hercules, CA, USA). For the reversal of resistance, cells were preincubated with ER or RSL3 for 1–2 h followed by the addition of various concentrations of drugs (or 1400W) and incubated for 24–72 h in the complete medium.

### 4.3. Accumulation of Adriamycin 

The accumulation of ADR in OVCAR-8 and NCI/ADR-RES cells was measured as described before by Shen et al. [38] with some modifications. Briefly, about 100,000 cells were seeded in six-well cover slips for 18 h in a complete medium. 1.0 mL of fresh medium was added, and cells were treated with different concentrations of ER, Verapamil (10 µM) or 1400W for 1 h. Then, ADR (0.1–1.0 µM) was added, and the cells were incubated for 1 h. The medium was removed, and the cells were washed in ice-cold PBS (pH 7.2), kept on ice in 1 mL of PBS, and examined using confocal microscopy. The average luminal fluorescence intensity of each cell was quantified using ImageJ software (Version 1.54). Around 20–25 images were taken for each treatment group and controls, and 20 ≤ cells per group were analyzed. The means of fluorescence intensity were analyzed using GraphPad Prism 8 (GraphPad Software, Inc, La Jolla, CA, USA). The data were subjected to one-way ANOVA.

### 4.4. Accumulations of Hoechst 33342 

Accumulations of Hoechst 33342 in MCF-7 and MCF-7/MX cells were carried out as described before by Galetti et al. [27] with some modifications. Briefly, about 1 × 10^5^ cells were seeded in six-well cover slips for 18 h in a complete medium. After 18 h, the medium was removed, and 2 mL of media was added, and cells were preincubated with different concentrations of ER for 1 h. Then Hoechst 33342 (5 × 10^−7^ M) was added, and the cells were incubated for 2 h. When used, 1400W was preincubated for 1 h. Cells were washed with ice-cold PBS (×2), kept on ice in 1 ml of PBS, and examined using confocal microscopy and analyzed as described above.

### 4.5. Western Blots 

Cell pellets following treatment with ER (2.5 µM) for 4 and 24 h were collected, washed (ice-cold PBS) and homogenized in 200µl PierceTM RIPA (Thermo Fisher, Carlsbad, CA, USA, 89900) supplemented with 25× Complete Protease Inhibitor, EDTA-free (Roche, 11836170001). Samples were centrifuged for 15 min at 14,000× *g* at 4 °C, and supernatant aliquoted and stored at −80 °C. The concentration of protein in the samples was determined using the PierceTM BCA Protein Assay (Thermo Fisher, 23225). Fifteen µg of protein was combined with a NuPage LDS Sample Buffer (Invitrogen, NP0007) and a NuPage Sample Reducing Agent (Invitrogen, Waltham, MA, USA NP 0009) and incubated at 70 °C for 10 min. Samples were loaded onto a NuPAGE™ 4 to 12%, Bis–Tris gel (Invitrogen, NP0336BOX), and run for 35 min at 200 V in 1× NuPage MES Run Buffer (Invitrogen, NP0002) to which NuPage Antioxidant (Invitrogen, NP0005) had been added. SeeBlue™ Plus2 Pre-stained Protein Standard Protein (ThermoFisher) was used as a reference for protein size. Proteins were transferred from the gel to nitrocellulose membranes using the iBlot™ Gel Transfer Device (ThermoFisher). Ponceau S solution (Sigma, St. Louis, MO, USA P7170) was used to visualize total protein transfer. EveryBlot Blocking Reagent (Bio-Rad, 120110020) was used to block membranes. Membranes were incubated with a primary antibody overnight at 4 °C and diluted in EveryBlot Blocking Reagent: rabbit anti-GADPH (Novus, NBP2-20859; 1:1000), mouse anti-Pgp (Santa Cruz Biotechnology, TX, USA; 1:200), and rabbit anti-BCRP (Abcam, 1:1000). Membranes were washed 3 × 5 min in 1× tris-buffered saline (TBS; Bio-Rad, Hercules, CA, USA, 1706435) plus 0.1% tween-20 (Sigma, St. Louis, MO, USA, P7949) (TBST). Membranes were incubated for 45 min at room temperature with secondary antibody diluted in EveryBlot Blocking Reagent (Bio-Rad): goat-anti-rabbit HRP (Novus, NB7160; 1:5000) or goat-anti-mouse HRP (Invitrogen, 32230; 1:1000). The excess secondary antibody was removed by washing 3 × 5 min in TBS. HRP was visualized using SuperSignal™ West Pico Plus Chemiluminescent Substrate (Thermo Scientific, Waltham, MA, USA, 34580).

### 4.6. Molecular Modeling 

Various ligand-bound ABCB1 and ABCG2 structures were selected for the docking studies. X-ray crystal structures 6QEE (Zosuquidar), 6QEX (Taxol), 7A69 (Vinsristin), 7A6C (Elacridar), 7A6E (Tariquidar), 7A6F (Zosuqudar), 7O9W (Encequidar), and 8Y6H (Elacridar) in which ligands were indicated within parenthesis were selected for ABCB1 docking. After removing the ligand, AutoDock Vina [39] (Scripps Research Institute, La Jolla, CA, USA) was used for docking with the inner box dimension of 30 Å × 30 Å × 30 Å where the box was centered at the coordinate center of the ligand found with the X-ray crystal structure. Residues within the box were assumed to have flexible side chains, and the exhaustiveness level was set to 32. For ABCG2, X-ray crystal structures 6VXH (Imanitib), 6VXI (Mitoxantrone), 6VXJ (SN-38), 7NEQ (Tariquidar), 7NEZ (Topotecan), 7NFD (Mitoxantrone), 7OJ8 (Estrone-3-sulfate), 7OJH (Topotecan, the protein was in turnover state 1), and 7OJI (Topotecan, the protein was in turnover state 2) were used. For ABCB1, ER, Tariquidar, Adriamycin, and Taxol were used as docked ligands while for ABCG2, docking ligands were ER, Camptothecin, Topotecan, Tariquidar, and Mitoxantrone.

### 4.7. Statistical Analysis 

The results are expressed as mean ± SEM of minimum of three independent experiments (*n* = 3). A one-way analysis of variance (ANOVA) or Student’s t-test was used for statistical analysis using Graph Pad Prism (GraphPad Software, Inc, La Jolla, CA, USA). For multiple comparisons, the Tukey Multiple comparison test was utilized, and results were considered statistically significant when *p* <0.05.

## 5. Conclusions

Our study demonstrates that the ferroptosis inducer, Erastin, exhibits strong cytotoxicity against ABCB1- and ABCG2-expressing tumor cells, indicating it is not a substrate for these ABC transporters in NCI/ADR-RES and MCF-7/MXR cells. Additionally, the GPX4 inhibitor, RSL3, shows even greater cytotoxicity in these cells. Notably, both Erastin and RSL3 significantly enhance the cytotoxic effects of Adriamycin and Topotecan in Pgp- and BCRP-expressing cells, with RSL3 proving more effective by also sensitizing drug-sensitive cells. The enhanced cytotoxicity of Adriamycin and Topotecan appears to result from increased drug uptake/retention due to inhibition of ABC transporter ATPase functions by ^●^NO or related species generated by Erastin. This effect is attenuated by a selective iNOS inhibitor. Further research is needed to elucidate the roles of iNOS, ^●^NO, and the regulation of ABC transporters in this context.

## Figures and Tables

**Figure 1 ijms-26-00635-f001:**
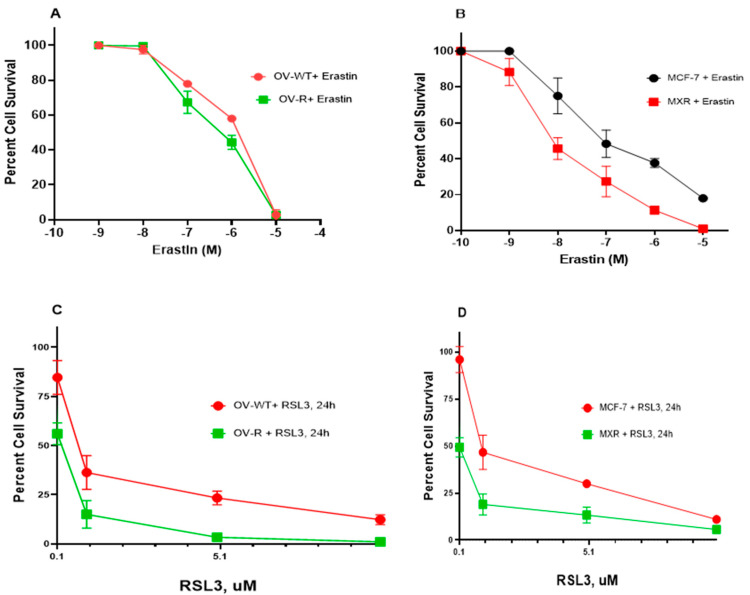
Cytotoxicity of ER (**A**,**B**) and RSL3 (**C**,**D**) in OVCAR-8 (OV-WT), NCI/ADR-RES (OV-R), MCF-7, and MCF7/MXR cells. Cytotoxicity studies were conducted as described in the methods section. ER cytotoxicity was determined following a 72 h drug treatment while RSL3 cytotoxicity was determined following a 24 h drug treatment.

**Figure 2 ijms-26-00635-f002:**
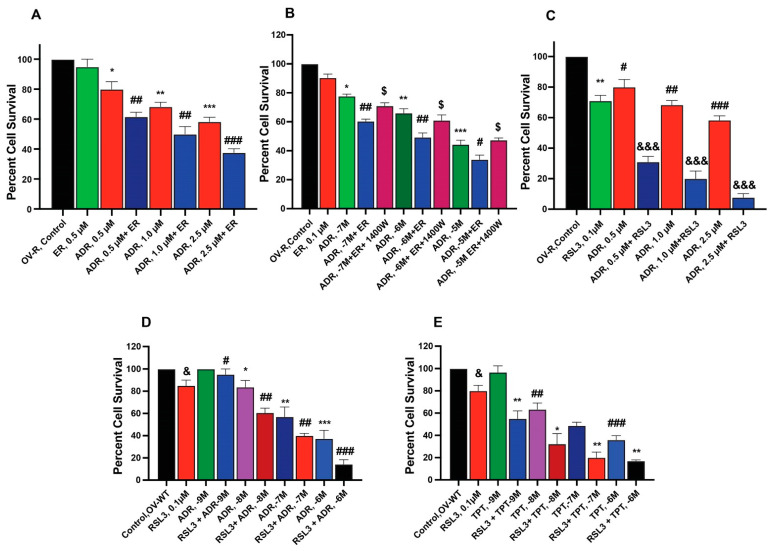
Sensitization of ADR (**A**) and TPT (**E**) by ER and RSL3 (**C**–**E**) and effects of 1400W (**B**) on ER-mediated cytotoxicity of ADR in ovarian tumor cells (OV-R and OV-WT). *, **, and *** are *p* values < 0.05, 0.005 and 0.001 against untreated controls. #, ##, and ### are *p* values < 0.05, 0.005 and 0.001, respectively, and $, &, and &&& are *p* values < 0.05, and 0.001, respectively, against matched samples of ADR- and TPT-treated samples alone.

**Figure 3 ijms-26-00635-f003:**
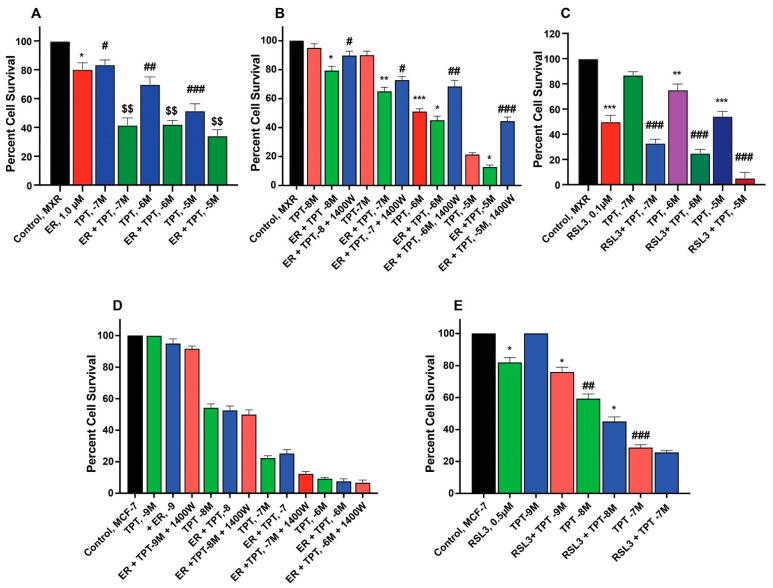
Sensitization of TPT by ER (**A**,**D**) and RSL3 (**C**,**E**) and effects of 1400W (**B**,**D**) on ER-mediated cytotoxicity of TPT in MCF-7 and MCF-7/MXR breast tumor cells. *, **, and *** are *p* values < 0.05, 0.005 and 0.001 against untreated controls. #, ##, and ### are *p* values < 0.05, 0.005 and 0.001, respectively, against matched samples of TPT-treated samples alone. $$ is *p* values < 0.005 against ER-treated control.

**Figure 4 ijms-26-00635-f004:**
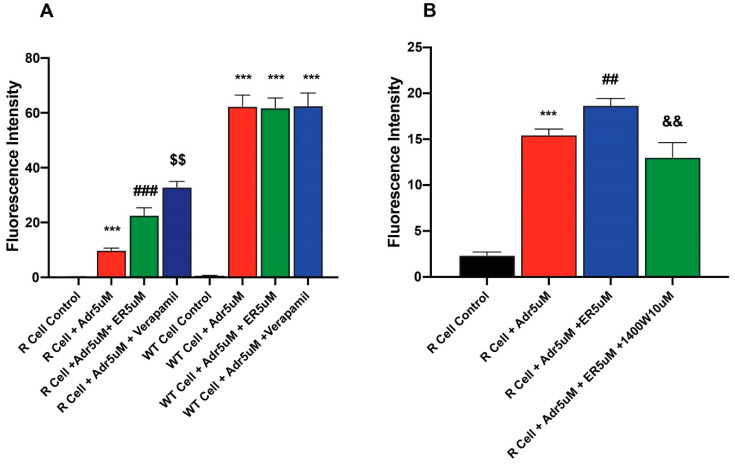
Effects of (**A**) ER on the uptake/retention of Adriamycin in OV-WT and NCI/ADR-RES cells and (**B**) effects of Verapamil and 1400W on ER-induced uptake of ADR. ***, *p* values < 0001 compared to the control. ###, *p* value< 0.001 compared to ADR alone. ## and $$ are *p* values < 0.005 compared to ER + ADR. && is *p* value < 0.005 against ADR + ER treated cells.

**Figure 5 ijms-26-00635-f005:**
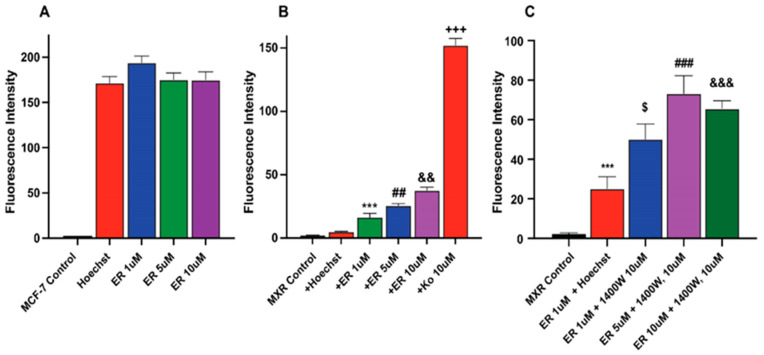
Uptake of Hoechst dye in (**A**) MCF-7 (**B**) MCF-7/MXR cells following treatment with different concentrations of ER and (**C**) effects of 1400W on ER-mediated uptake of the dye in MCF-7/MXR cells. ***, *p* value < 0.001 compared to untreated control. $, *p* value < 0.05 compared to ER 1.0 µM alone. ##, ###, *p* value < 0.005 and <0.001 compared to untreated control and ER 5.0 µM alone, respectively, and &&, &&&, *p* value < 0.005 and <0.001 compared to ER, 5 µM, and 10 µM, respectively and +++ *p* values < 0.001 compared to ER, 5 µM alone.

**Figure 6 ijms-26-00635-f006:**
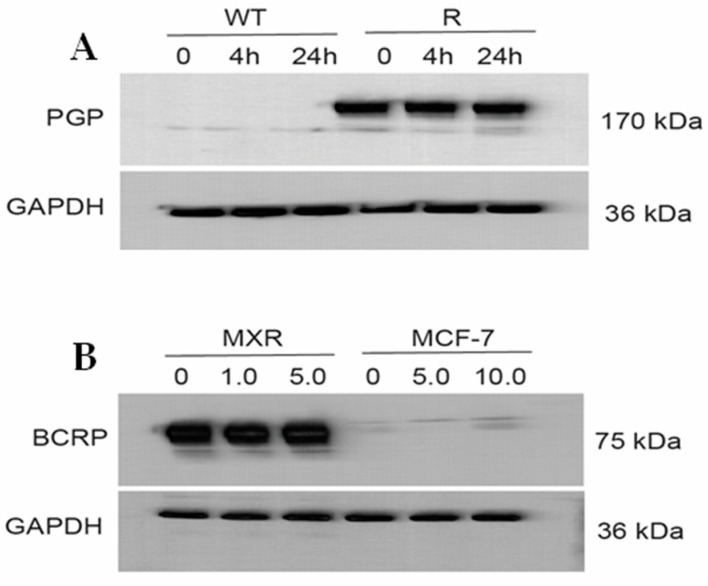
Western blots (**A**) for P-170 in OVCAR-8 (WT) and NCI/ADR-RES (R) cells and (**B**) for BCRP in MCF-7/MXR and MCF-7 cells following treatment with different concentrations of Erastin for 4 h or 24 h.

**Figure 7 ijms-26-00635-f007:**
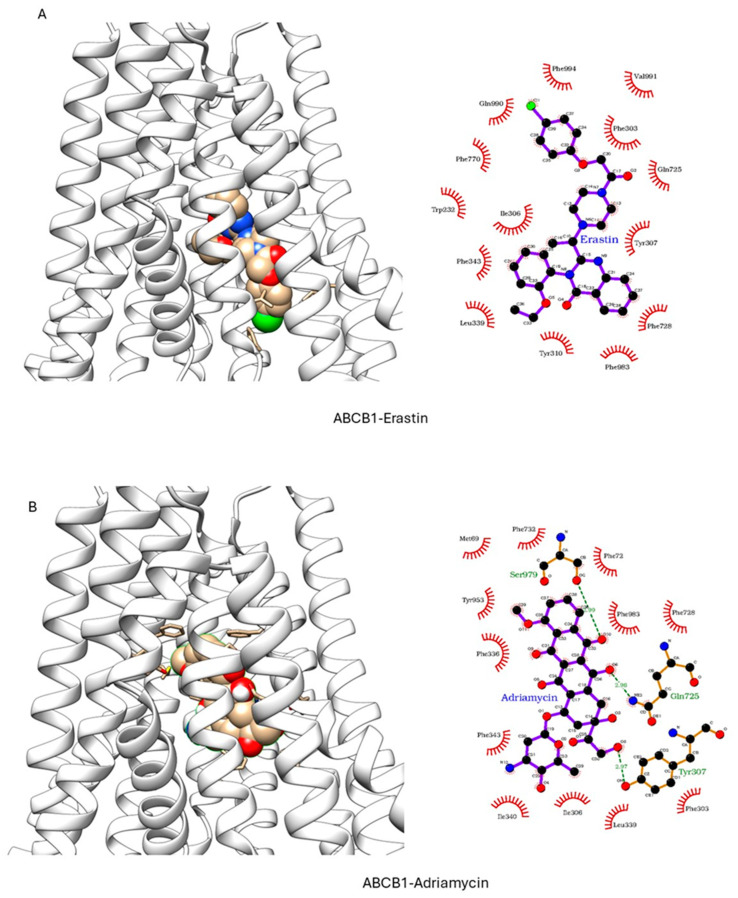
Docking of (**A**) Erastin and (**B**) Adriamycin in P-gp (ABCB1, pdb ID: 6C0V.pdb) ER and ADR are shown in solid spheres, and the proteins are represented by ribbon diagrams. Residues of the protein that are in contact or forming H-bonds are shown in righthand panels.

**Figure 8 ijms-26-00635-f008:**
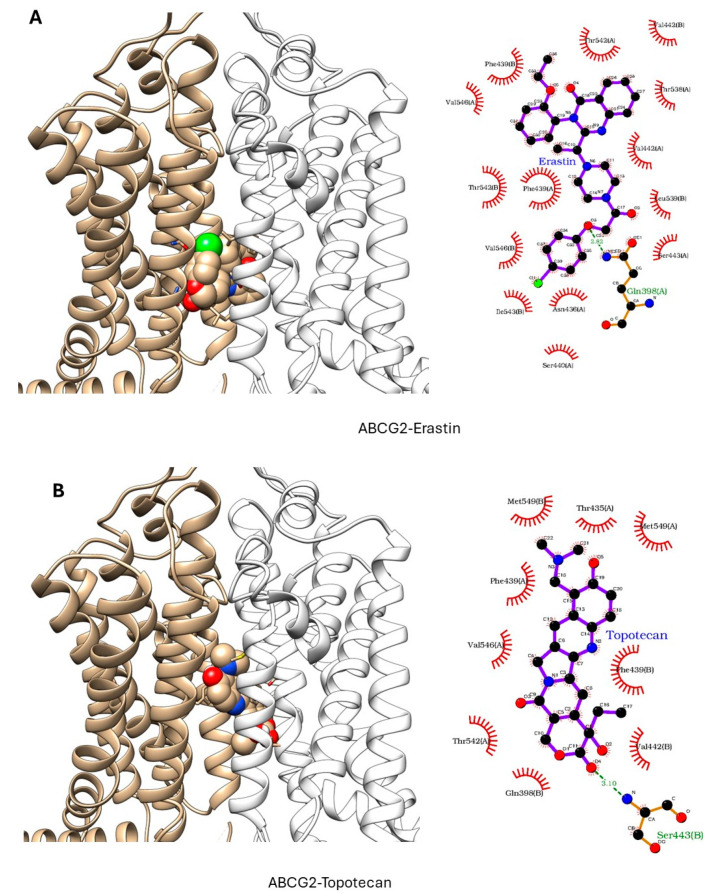
Docking of (**A**) Erastin and (**B**) Topotecan in BCRP (ABCG2, pdb ID: 6hbu.pdb). ER and TPT are shown in solid spheres, and the proteins are represented by ribbon diagrams. Residues of the protein that are in contact or forming H-bonds are shown in righthand panels.

**Table 1 ijms-26-00635-t001:** Inhibitor affinities for ABCB1 (P-gp; in kcal/mol). PDB identities with the ligand used in the crystallization are given in the column headings, and the ligands used in the docking are given in row headings.

Drugs	6QEE Zosuquidar	6QEX Taxol	7A69 Vincristine	7A6C Elacridar	7A6E Tariquidar	7A6F Zosuquidar	7O9W Encequidar	8Y6H Elacridar
Erastin	−9.6	−9.9	−9.5	−10.0	−10.5	−9.5	−9.8	−10.4
Tariquidar	−10.7	−10.8	−9.7	−11.3	−11.5	−10.4	−11.4	−13.0
Adriamycin	−9.0	−9.4	−9.0	−9.6	−10.2	−8.6	−9.3	−10.6
Taxol	−10.3	−10.3	−10.2	−11.7	−11.0	−11.0	−10.4	−11.4

**Table 2 ijms-26-00635-t002:** Inhibitor affinities for ABCG2 (BCRP; in kcal/mol). PDB identities with the ligand used in the crystallization are given in the column headings, and the ligands used in the docking are given in row headings.

Drugs	6VXH Imatinib	6VXI Mitoxantrone	6VXJ SN38	7NEQ Tariquidar	7NEZ Topotecan	7NFD Mitoxantrone	7OJ8 Estrone 3-sulfonate	7OJH Topotecan TOstate-1	7OJH Topotecan TOState-2
Erastin	−9.1	−9.5	−9.1	−9.8	−8.8	−8.6	−9.0	−9.3	−9.7
Camptothecin	−9.1	−11.0	−10.9	−10.8	−10.0	−10.2	−10.8	−10.4	−11.1
Topotecan	−8.6	−10.6	−10.1	−10.8	−9.8	−9.9	−10.8	−10.3	−10.6
Tariquidar	−11.0	−11.9	−11.6	−11.3	−11.3	−10.3	−11.6	−11.6	−12.4
Mitoxantrone	−7.1	−8.3	−8.2	−7.6	−7.0	−8.0	−8.0	−7.4	−8.4

## Data Availability

The original contributions presented in this study are included in the article/Appendix A. Further inquiries can be directed to the corresponding author(s).

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
