# Peer review of "Ferroptosis Inducers Erastin and RSL3 Enhance Adriamycin and Topotecan Sensitivity in ABCB1/ABCG2-Expressing Tumor Cells"

_ijms, 2025, doi:10.3390/ijms26020635_

Round 1

Reviewer 1 Report

Comments and Suggestions for Authors

Birandra et al. submitted the manuscript entitled: Ferroptosis Inducers Erastin and RSL3 Enhance Adriamycin and Topotecan Sensitivity in ABCB1/ABCG2-Expressing Tumor cells, in which the authors reported the sensitization of ferroptosis Inducers against drug-resistant tumor cells. The authors co-incubated Erastin or RSL3 with ADR or TPT and found a non-toxic dose of ferroptosis inducers can reverse the drug-resistant profile of tumor cells. For primary mechanism investigation, the authors found that the ER pretreatment can lead to ADR accumulation and in the same time, enhanced uptake of Hoechst Dye, while the expression of P-gp or BCRP remain unaffected. Generally, this is a well-prepared work and the topic will be of interest to potential readers of IJMS.

I have some minor comments as follows.

1. The authors did not seem to profile the ferroptosis phenotype in this work. In order to correlate observed sensitization with ferroptosis, the authors are encouraged to include more experiments eg. lipid peroxidation assay. This result can also help rule out the possibility of non-ferroptosis pharmacology induced by RSL3.

2. Figure 4: 5 uM of ER should be toxic to bothOV-WT and NCI/ADR-Res cells. How to rule out the possibility that this intake is not cell-death related?

3. I tend to believe that the function of ER-induced sensitization is not directly related to P-gp, hence the result in figure 6 does not really surprise me. However, the author did observe higher uptake when treated with ER. Have the authors tried other P-gp inhibitor to see if they can also induce Hoechst uptake?

Author Response

I have some minor comments as follows.

  1. The authors did not seem to profile the ferroptosis phenotype in this work. In order to correlate observed sensitization with ferroptosis, the authors are encouraged to include more experiments eg. lipid peroxidation assay. This result can also help rule out the possibility of non-ferroptosis pharmacology induced by RSL3.

We have recently published about the mechanism of cell death induced by Erastin in these cell lines, Sinha et al., IJMS, 2024. In this manuscript we have described inhibition of Xc transport and lipid peroxidation induced by Erastin. RSL3 significantly enhanced effects of Erastin.

  1. Figure 4: 5 uM of ER should be toxic to bothOV-WT and NCI/ADR-Res cells. How to rule out the possibility that this intake is not cell-death related?

Because these are very short- incubations, e.g., 1-2 h, little or no toxicity is expected or observed.

  1. I tend to believe that the function of ER-induced sensitizationis not directly related to P-gp, hence the result in figure 6 does not really surprise me. However, the author did observe higher uptake when treated with ER. Have the authors tried other P-gp inhibitor to see if they can also induce Hoechst uptake?

No, unfortunately we did not utilize any other inhibitors in this assay.

Reviewer 2 Report

Comments and Suggestions for Authors

In the study, "Ferroptosis Inducers Erastin and RSL3 Enhance Adriamycin and Topotecan Sensitivity in ABCB1/ABCG2-Expressing Tumor Cells", the authors investigated the potential of Erastin and RSL3 to overcome chemoresistance in cancer cells that overexpress ABC transporters. The researchers discovered that these inducers augmented the cytotoxicity of Adriamycin and topotecan by enhancing drug uptake in cells that overexpress P-gp and BCRP. The impact of Erastin was neutralized by 1400W, indicating that iNOS plays a role in its mechanism of action. While the paper provides valuable data on ferroptosis inducers, its innovation is relatively limited, as it does not introduce breakthrough concepts, new technologies, or deep clinical translation research, several concerns require further investigation.

Major concerns:

1. The study employs two cell lines, NCI/ADR-RES and MCF-7/MXR, which exhibit overexpression of P-gp and BCRP in ovarian and breast cancer. However, it should be noted that the aforementioned cell lines may not fully reflect the MDR mechanisms observed in other ABCB1/ABCG2-expressing tumor types. Therefore, the applicability of the findings may be limited to these specific models, and further testing in other tumor types is necessary to assess the broader relevance of the results.

2. This study is primarily concerned with in vitro results obtained using cell lines. However, to more accurately assess the in vivo effects, it would be beneficial to consider incorporating data from animal models and clinical samples.

3. The current study relies on a single method to measure drug cytotoxicity. It is recommended to incorporate additional cytotoxicity assays to provide more comprehensive data and further validate the reliability and accuracy of the results.

4. To enhance the clarity and precision of the experimental results, it is necessary to include fluorescence images obtained from the Accumulation of Adriamycin and Accumulation of Hoechst 33342 experiments conducted using confocal microscopy.

Minor concerns:

1. Statistical analysis is missing in several figures.

2. The statistical symbols utilized in the figure are excessive and detract from the clarity of the illustration. It is advised that their usage be simplified and that only the essential annotations be retained in order to enhance the clarity and readability of the figure.

3. The font in line 237 is incorrect. It is recommended to adjust it to match the font style.

4. There are some typos throughout the text.

Author Response

Reviewer-2Top of Form

In the study, "Ferroptosis Inducers Erastin and RSL3 Enhance Adriamycin and Topotecan Sensitivity in ABCB1/ABCG2-Expressing Tumor Cells", the authors investigated the potential of Erastin and RSL3 to overcome chemoresistance in cancer cells that overexpress ABC transporters. The researchers discovered that these inducers augmented the cytotoxicity of Adriamycin and topotecan by enhancing drug uptake in cells that overexpress P-gp and BCRP. The impact of Erastin was neutralized by 1400W, indicating that iNOS plays a role in its mechanism of action. While the paper provides valuable data on ferroptosis inducers, its innovation is relatively limited, as it does not introduce breakthrough concepts, new technologies, or deep clinical translation research, several concerns require further investigation.

We recognize that our study focuses primarily on building a robust dataset on ferroptosis inducers rather than introducing breakthrough technologies or clinical applications. Our goal was to provide foundational data to facilitate future research and translational efforts. Incremental advances like ours often lay the groundwork for significant innovations. While the innovation of our study may appear limited, we believe that the comprehensive profiling of ferroptosis inducers and their mechanisms of action provides a valuable resource for researchers aiming to bridge the gap between basic science and clinical application.

Major concerns:

  1. The study employs two cell lines, NCI/ADR-RES and MCF-7/MXR, which exhibit overexpression of P-gp and BCRP in ovarian and breast cancer. However, it should be noted that the aforementioned cell lines may not fully reflect the MDR mechanisms observed in other ABCB1/ABCG2-expressing tumor types. Therefore, the applicability of the findings may be limited to these specific models, and further testing in other tumor types is necessary to assess the broader relevance of the results.
  2. This study is primarily concerned with in vitro results obtained using cell lines. However, to more accurately assess the in vivo effects, it would be beneficial to consider incorporating data from animal models and clinical samples.

I believe the comments regarding the cell lines used in these studies do not accurately reflect their significance or utility. These cell lines have been extensively used to study the role of multidrug resistance (MDR) in tumor biology and treatment. They have been featured in many of our publications without raising such criticism from other reviewers. Furthermore, prominent groups, including the Gottesman lab at the NCI, have frequently employed the NCI/ADR-RES line. These cell lines exhibit all the key characteristics of MDR models. While it is true that MDR cell lines differ based on the selecting agents used, they nonetheless provide valuable insights, as demonstrated in this study.

Regarding the suggestion for in vivo studies, we have already indicated in our manuscript that these experiments will be conducted in future studies. That said, I have often questioned the relevance of such studies in mice to human biology, considering the considerable resources, time, manpower, and funding they require.

  1. The current study relies on a single method to measure drug cytotoxicity. It is recommended to incorporate additional cytotoxicity assays to provide more comprehensive data and further validate the reliability and accuracy of the results.

The concern regarding cytotoxicity variability is also misplaced. Cytotoxicity depends on factors such as exposure time and cell density, which determine the effective drug concentration. Consistency in experimental parameters—such as drug concentration and exposure time—ensures reliable results. While some methods may be more sensitive than others due to number of cells utilized, we used two robust methods, namely CellTiter-Glo and cell counting, to confirm our findings.

  1. To enhance the clarity and precision of the experimental results, it is necessary to include fluorescence images obtained from the Accumulation of Adriamycin and Accumulation of Hoechst 33342 experiments conducted using confocal microscopy.

Finally, the suggestion to include additional images does not necessarily enhance clarity. Given the sheer number of panels already included (4–5 panels per figure, with a total of 12–18 images), adding more would make the presentation overly complex.

However, we have supplemented our manuscript with fluorescence images of Adriamycin and Hoechst uptake as supplementary figures, 2 and 3.

Minor concerns:

  1. Statistical analysis is missing in several figures.

The only figure that had missing asterisk was Figure-5B which has been redone to include this in the revised manuscript.

  1. The statistical symbols utilized in the figure are excessive and detract from the clarity of the illustration. It is advised that their usage be simplified and that only the essential annotations be retained in order to enhance the clarity and readability of the figure.

We believe these are essential for statical analysis.

  1. The font in line 237 is incorrect. It is recommended to adjust it to match the font style.

Problems in type setting by the publishers.

  1. There are some typos throughout the text.

Has been corrected.